# Prognostic models for predicting in-hospital paediatric mortality in resource-limited countries: a systematic review

Morris Ogero ,[1,2] Rachel Jelagat Sarguta,[1] Lucas Malla,[2] Jalemba Aluvaala,[2] Ambrose Agweyu,[2] Mike English,[2,3] Nelson Owuor Onyango,[1] Samuel Akech[2]

¹School of Mathematics, University of Nairobi College of Biological and Physical Sciences, Nairobi, Kenya
²Health Services Unit, KEMRI-Wellcome Trust Research Programme, Nairobi, Kenya
³Nuffield Department of Medicine and Department of Paediatrics, Oxford University, Oxford, UK

**Correspondence to**
Morris Ogero;
mogero@kemri-wellcome.org

## ABSTRACT

**Objectives** To identify and appraise the methodological rigour of multivariable prognostic models predicting in-hospital paediatric mortality in low-income and middle-income countries (LMICs).

**Design** Systematic review of peer-reviewed journals.

**Data sources** MEDLINE, CINAHL, Google Scholar and Web of Science electronic databases since inception to August 2019.

**Eligibility criteria** We included model development studies predicting in-hospital paediatric mortality in LMIC.

**Data extraction and synthesis** This systematic review followed the Checklist for critical Appraisal and data extraction for systematic Reviews of prediction Modelling Studies framework. The risk of bias assessment was conducted using Prediction model Risk of Bias Assessment Tool (PROBAST). No quantitative summary was conducted due to substantial heterogeneity that was observed after assessing the studies included.

**Results** Our search strategy identified a total of 4054 unique articles. Among these, 3545 articles were excluded after review of titles and abstracts as they covered non-relevant topics. Full texts of 509 articles were screened for eligibility, of which 15 studies reporting 21 models met the eligibility criteria. Based on the PROBAST tool, risk of bias was assessed in four domains; participant, predictors, outcome and analyses. The domain of statistical analyses was the main area of concern where none of the included models was judged to be of low risk of bias.

**Conclusion** This review identified 21 models predicting in-hospital paediatric mortality in LMIC. However, most reports characterising these models are of poor quality when judged against recent reporting standards due to a high risk of bias. Future studies should adhere to standardised methodological criteria and progress from identifying new risk scores to validating or adapting existing scores.

**PROSPERO registration number** CRD42018088599.

## INTRODUCTION

Over recent decades, there has been considerable progress in improving child survival[1] but child mortality remains high in sub-Saharan Africa relative to the rest of the world.[2] Paediatric deaths in hospitalised children mostly occur soon after admission,[3] and are caused by common conditions such as malaria, pneumonia and diarrhoeal diseases among others, which are readily treatable by cost-effective interventions.[3–5] In low-income and middle-income countries (LMICs), clinicians often use a set of clinical signs as recommended in the guidelines by WHO to identify patients at risk of deterioration while making decisions on appropriate treatment.[6] Clinical criteria recommended by WHO were developed following expert recommendations based on review of evidence from studies reporting risk factors for mortality. Prognostic/predictive models use statistical equations to predict high-risk patients based on the combination of risk factors. Use of these models by clinicians may improve patients' outcomes by enhancing clinicians' ability in identifying patients at the risk of deterioration.[7]

Several prognostic models for hospitalised children have been published over the last three decades,[8] however, there are doubts as to whether authors of these models used the appropriate methodology in their development.[9] Notably, in the current clinical practice guidelines, none of these models have been recommended for use in resource-limited

setting and reviews of the methodology used in their development have been highly recommended.[10] This systematic review addresses this need and aims at identifying and summarising existing studies reporting prognostic models or scoring systems predicting in-hospital paediatric mortality in LMIC. Specifically, the research summarises the evidence from the published studies and appraises the methodological rigour of each existing model.

## Strengths and limitations of this study

► This is the first review on methodological rigour of models predicting paediatric mortality in resource-limited settings.
► We used a robust search strategy with no language restriction yielding many potentially eligible studies.
► Due to substantial heterogeneity in the models included no meta-analyses was conducted.
► We relied on what was reported to determine the risk of bias in prognostic models included.
► Google Translate was used to translate one study from French to English.

## METHODS

### Protocol and registration

As recommended, a research protocol for this review was published in a peer-reviewed journal.[11] This study is reported as per guidelines by the Preferred Reporting Items for Systematic Reviews and Meta-Analyses.[12]

### Eligibility criteria

We used the following eligibility criteria for inclusion of articles:

1. Study design: we included peer-reviewed studies whose study design was either a case–control, cohort (prospective or retrospective), cross-sectional or randomised controlled trial.
2. Outcome: we included studies predicting all-cause in-hospital mortality. Studies predicting operative, trauma or postdischarge mortality were excluded.
3. Setting and target population: we focused on studies targeting over 1-month-old children admitted in paediatric wards within resource-limited settings as specified by the World Bank.[13] Studies whose target population were children in high-dependency unit or intensive care unit were excluded because of limited availability of such facilities in LMIC. We also excluded studies whose target population included conditions not common in children, such as diabetes, cancer, chronic kidney disease, musculoskeletal disorders. However, if a study focused on one of the common childhood illnesses such as malaria, pneumonia, meningitis, anaemia and diarrhoea/dehydration,[3] then it was included.
4. Prognostic research studies: we included studies whose main objective was deriving a predictive model(s) or scoring system(s). We excluded case-series, conference proceedings, editorials, commentaries, expert views, case reports, reviews and studies that mainly generate hypothesis such as explanatory studies.[14]
5. Predictors in the model: studies that reported multivariable model with at least two variables/predictors were included.
6. Full text and language: no language restrictions were made, we used Google Translate to translate non-English language studies. We excluded studies that were not available in full text.

### Search strategy of articles

Based on Checklist for critical Appraisal and data extraction for systematic Reviews of prediction Modelling

Studies (CHARMS) checklist,[15] we identified seven core items (see online supplemental table 1) specific to our study that guided the formulation of the eligibility criteria, review aims and the search strategy.

Where applicable, Medical Subject Headlines terms and keywords were used to identify research papers developing predictive models relevant for this review (see online supplemental table 2). We conducted a search of articles in CINAHL (via EbscoHost), Google Scholar, MEDLINE and Web of Science published since inception to August 2019. To identify other potentially eligible studies, we manually searched reference lists of the identified articles and collated the final search results in EndNoteX7 bibliography tool.

### Screening of articles for inclusion

Prior to screening titles and abstracts, two reviewers (MO and LM) standardised the approach to be used in the process of screening and a sample of 30 articles were used to familiarise and train reviewers (MO, LM and JA) on the process of screening of articles and data abstraction. Two reviewers (MO and LM) screened articles' titles and abstracts. Disagreements were resolved via discussion and the third reviewer (JA) adjudicated the final decision where necessary.

### Data extraction from the included articles

For each of the study included, we used CHARMS guidelines to abstract the following data items; participant enrolment, study design, study population characteristics, location, sample size, number and selection of predictors, study dates, handling of continuous predictors, missing data, method of modelling (eg, logistic regression or survival), verification of model assumptions, internal validation methods (eg, resampling techniques such as cross validations and bootstrapping, or random split of data); model presentation (eg, full regression formula with coefficients, score chart or nomogram); and model performance metrics including discrimination—area under the curve (AUC) accompanied with 95% CI; calibration; classification metrics including specificity, sensitivity, positive and negative predictive values. We further explored literature to determine if included models have been externally validated elsewhere. We treated each model separately for articles that developed multiple prognostic models. Data extracted from articles by the two reviewers (MO and LM) were compared and disagreements were resolved via discussion with the third reviewer (JA). Due to substantial heterogeneity that was observed after assessing studies included, we did not conduct a quantitative summary of the identified models.

### Assessment of methodological rigor of the identified prognostic models

Based on Prediction study Risk Of Bias Assessment Tool (PROBAST) a Cochrane tool for assessing risk of bias (RoB) in predictive models,[16 17] we assessed the RoB for each model in four domains: selection of the study

participant, predictors domain (eg, selection of the candidate predictors), statistical analysis domain (eg, sample size, continuous predictors, missing data) and outcome domain. See online supplemental table 3 for details. In each domain, there were a set of signalling questions each with five possible answers: yes; probably yes; probably no; no and no information. Any positive answer (yes or probably yes) suggests low RoB. There were three possible outcomes per domain namely: low; high or unclear RoB. Using these outcomes, we came up with an overall rating of RoB for each model. As recommended by PROBAST, a prognostic model was rated to be of 'low RoB' if all four domains had an outcome of 'low'. A prognostic model was rated 'high RoB' if at least one domain had an outcome of 'high'. Finally, a prognostic model was rated as 'unclear RoB' if at least one domain had an outcome of 'unclear' and the rest of the domains had an outcome of 'low'.

## Patient and public involvement
No patient or public involvement.

## RESULTS
### Characteristics of the included studies
Our search strategy identified a total of 4054 unique articles, 3545 articles were excluded after review of titles and abstracts as they reported non-relevant topics. Full texts of 509 articles were assessed for eligibility, of which 15 primary studies reporting 21 developed models met the eligibility criteria (figure 1). The eligible studies analysed data for patients who were below 15 years of age with median mortality being 6.7% (range 1.2%–43.9%).[18 19] While majority of the models were developed for general cases in paediatric wards (n=9), some were tailored for specific paediatric groups defined by common diagnoses such as febrile illness (n=1),[20] malaria (n=2),[21 22] pneumonia (n=4),[18 23–25] malnutrition (n=2)[26 27] and other infectious diseases (n=3) (see online supplemental file 2).

Most of the included studies have been published post year 2000 (n=20) except for one study[26] published in 1996. The latest data used in the models under review were from 2016 to 2017 by Rosman et al[28] and the oldest data were used by Dramaix et al[26] from 1986 to 1988.

Five reports of the 15 included studies used data from at least two hospitals of which three studies[20 21 25] were conducted in multiple countries including sub-Saharan Africa and Asian countries (figure 2). Of the reviewed studies, most of the information we were abstracting were either not reported or were partially reported, an indication of non-adherence to the Transparent Reporting of a Multivariable Prognostic Model for Individual Prognosis or Diagnosis guidelines (TRIPOD).[29 30]

### Summary of issues in methodology of the reviewed models
#### Candidate predictors
There were 61 distinct predictors used in the final reported models with a median of 7 predictors in any one model. Initial selection of the independent candidate

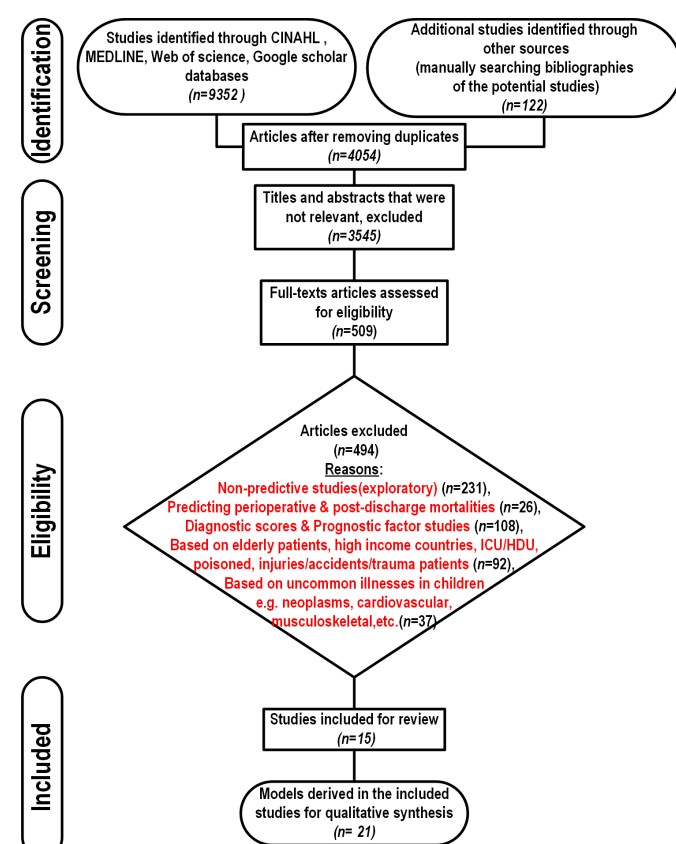

**Figure 1** PRISMA flow diagram showing the process used to identify prognostic models predicting in-hospital paediatric mortality included in this review. HDU, high-dependency unit; ICU, intensive care unit; PRISMA, Preferred Reporting Items for Systematic Reviews and Meta-Analyses.

predictors was mostly based on univariable analyses except for three studies[20 24 26] where the selection was based on literature reviews or clinical relevance. Backward stepwise selection method was used in six models in a multivariable analysis to determine final model predictors. Commonly included predictors in the final models included altered consciousness, malnutrition indicators, vital signs and signs of respiratory distress (see figure 3). Some models included predictors that were either not easy to obtain or required laboratory techniques. Of the 13 models that used continuous predictors, eight models categorised these continuous predictors where a continuous scale would have been possible. Two out of 13 models applied other techniques such as fractional polynomial[20] and restricted cubic splines[27] to determine the suitable functional form of the continuous predictors (see online supplemental file 2).

### Sample size, events per variable and missing data
Sample size ranged from 168[28] to 50249[31] with a median of 1307. The median events per variable (EPV) was 21 (IQR 8.3–32.5) of which seven models had less than 10 EPVs, suggestive of insufficient sample sizes which is prone to overfitting. For instance, 60 deaths were reported in the dataset used to develop Paediatric Early Death Index for

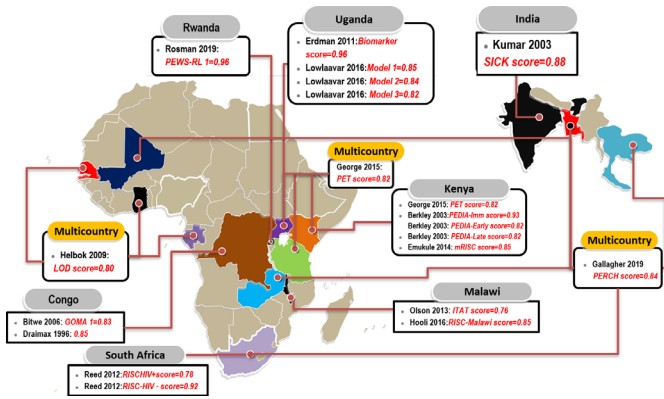

**Figure 2** Prognostic models predicting in-hospital paediatric mortality identified by country. Text highlighted in red are the names of the models with their corresponding discrimination measures (area under the curve). CRT, classification and regression trees; ITAT, inpatient triage assessment and treatmentscore; LOD, lambarene organ dysfunction; mRISC, modified Respiratory Index of Severity in Children; PEDIA, Paediatric Early Death Index; PERCH, Pneumonia Etiology Research for Child Health; PET, paediatric emergency triage; PEWS-RL, paediatric early warning score for resource-limited settings; SICK, Signs of Inflammation in Children that Kill.

Africa (PEDIA-Immediate score in the study by Berkley *et al*. In reference to the rule that a study developing a predictive model should have a minimum of 10 events (deaths) for each independent candidate predictor in a predictive model,[32] a model with a maximum of six predictors should have been considered but 10 predictors were considered instead hence making EPV to be 6.

Proportions of missing data was not always reported. Handling of missing data varied across the reviewed studies as follows: six models did not report handling of missing data; eight used complete case analysis (CCA); four used multiple imputations by chained equations; and one study[27] used single imputation.

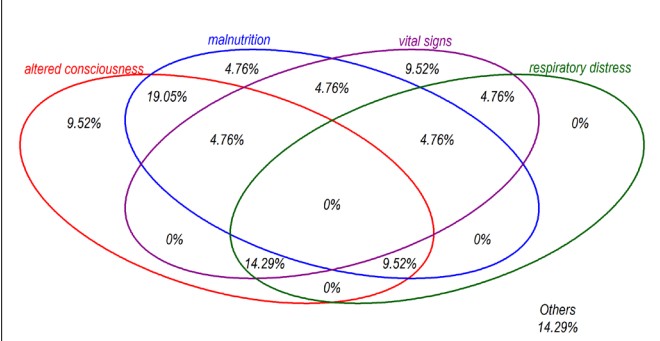

**Figure 3** Top four categories of predictors in the models of the reviewed reports: altered consciousness (coma, prostration, not alert, unconscious); malnutrition indicators (kwashiorkor, oedema, weight-for-height z-score, weight-for-age z-score, mid-upper arm circumference, wasting); vital signs (temperature, respiratory rate, heart rate, oxygen saturation); signs of respiratory distress (indrawing, lung crepitation, difficult breathing, grunting).

## Model development

Majority of the studies applied logistic regression, one study[20] used Cox regression, one study[19] used Spiegelhalter/Knill-Jones method, and another study[22] used a machine learning technique (classification and regression trees) in model development. Verification of model assumptions was not reported in most of the studies. For instance, George *et al*[20] despite using Cox regression to develop their model, did not report the verification of proportional hazard assumption nor explore the possibility of competing risks as recommended.[33] Other regression assumptions, for example, multicollinearity was equally not reported. However, since backward elimination method disregards redundant variables, we inferred the satisfaction of multicollinearity assumption if this method was applied.[34] Five studies developed models using data from different countries/centres but none of them clustered their analysis by source of data in a multilevel model to account for heterogeneity. Ignoring clustering leads to a biased predictor effect.[35]

## Model performance evaluation and presentation

Performance measures (both calibration and discrimination) were poorly reported in most of the studies and in most cases (n=20) AUC for discrimination was reported. Performance of the derived models was evaluated in 12 models using either split-sample, resampling methods or separate datasets. Except for the model derived by George *et al*[20] all other models did not report both apparent discrimination (without any adjustment for optimism) and optimism-corrected discrimination measures. Despite inadequate reporting of the models' performance, 16 models reported AUCs ≥0.80, an indication of promising models. Apart from the following exceptions; Lambarene Organ Dysfunction score,[21] PEDIA score,[19] Signs of Inflammation in Children that Kill (SICK) score,[36] Respiratory Index of Severity in Children (RISC) score[18] and modified RISC score,[23] other prognostic models in this review have not been externally validated (by independent investigators using diverse populations). Only two studies[24 37] developing four models provided a full model formula (both coefficients and intercept/baseline function) in their results as recommended.[29 30] While most of the models (n=17) were presented as simplified integer scores, only a few were assigned weights according to the regression coefficients.

## Risk of bias

Based on the PROBAST tool, RoB was assessed in four domains; participants, predictors, outcome and analyses. Figure 4 summarises the RoB assessment across all models included in this review where the domain of outcome was deemed to be of low RoB in all models. The domain of statistical analyses was the main area of concern where 19 out of 21 models did not report comprehensive details of model development as expected to warrant a proper RoB assessment using the nine signalling questions under the analyses domain. As a result, these models were judged

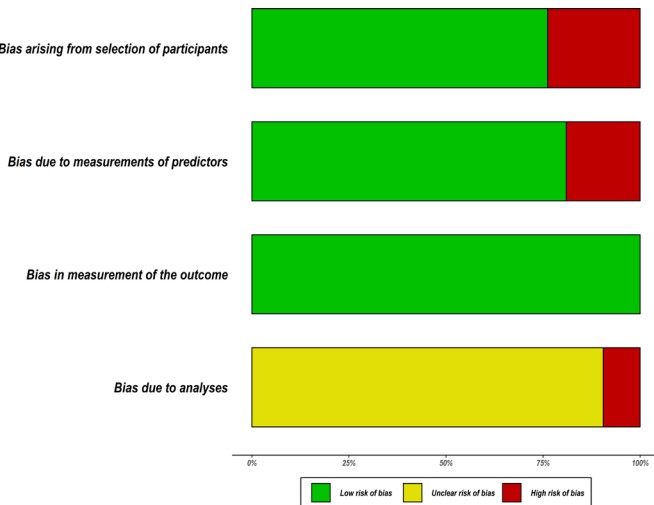

**Figure 4** Summary of the risk of bias of the included models using Prediction model Risk of Bias Assessment Tool.

to be of unclear RoB under this domain (see figure 5). Details on how models were scored against each of the PROBAST criterion (20 signalling questions) across the four domains are provided in the online supplemental

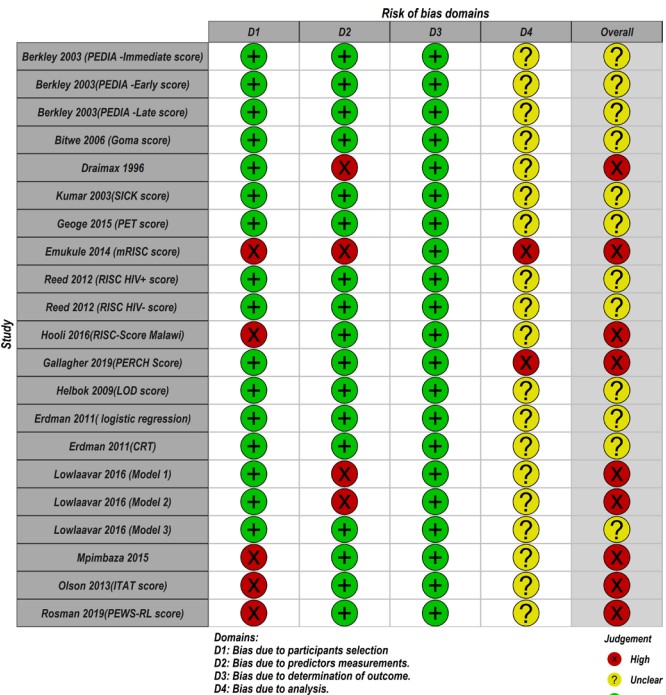

**Figure 5** Risk of bias assessment. Low means low risk of bias, high means a high risk of bias and unclear bias means it was not possible to assess the risk of bias. CRT, classification and regression trees; ITAT, inpatient triage assessment and treatment; LOD, lambarene organ dysfunction; mRISC, modified Respiratory Index of Severity in Children; PEDIA, Paediatric Early Death Index for Africa; PERCH, Pneumonia Etiology Research for Child Health; PET, paediatric emergency triage; PEWS-RL, Paediatric Early Warning Score for Resource-Limited; RISC, Respiratory Index of Severity in Children; SICK, Signs of Inflammation in Children that Kill.

file 3. In the overall judgement of RoB, 9 out of 21 models were judged to be of high RoB because at least one out of four domains in these models were rated as high RoB. The remaining models (12/21) were judged to be of unclear RoB on account of being rated low and unclear RoB in the domains. No model was rated low RoB in all four domains.

## DISCUSSION
### Summary of key findings
We conducted a systematic review to identify scores predicting in-hospital mortality for paediatrics in resource-limited countries. Fifteen studies that described the development of 21 prognostic models were identified. We describe characteristics of these studies as well as the methodological quality of the included models by using agreed recent guidelines applicable to predictive models. We have identified several important quality deficiencies such as inadequate reporting and other methodological concerns, including poor handling of missing data, automated selection of predictors, categorisation of continuous predictors, inadequate EPV and the poor presentation of the proposed model for use. As a result, no model was found to be of good methodological quality and consequently judged to be potentially high or unclear RoB in predictions (figure 5).

Our findings suggest that predictive models fail to meet recently agreed methodological criteria in various ways. First, in this review we observed that univariable analysis was routinely used in 18 out of 21 models in the selection of candidate predictors to be used in a multivariable analysis. This strategy tends to leave out possibly important prognostic factors which might be insignificant in a univariable analysis but turn out to be significant when combined with other predictors.[29 30] A priori selection of predictors using expert opinion, clinical intuition or literature is recommended for this purpose,[38 39] however, only three studies in this review employed this approach.[20 24 26]

Small sample sizes in model development can lead to poor predictive performance, overfitting and biased effect estimates. Prognostic models must have a minimum of 10 events per candidate independent predictor, as this is the accepted norm[40–42] and underpowered models arising from inadequate EPV increases the possibility of spurious results.[32] In this review, 7/21 models had inadequate sample sizes (EPV<10) and there was no information on whether bootstrapping, which serves to reduce overfitting was used in these models.[43]

Just like most of the epidemiological studies, missing data is a common problem which is solved using multiple imputation or other appropriate approaches, but this was rarely the case in the model development studies under this review. For instance, 8/21 models used CCA, 4/21 used multiple imputation under the missing at random (MAR) assumption and 6/21models did not report handling of missing data and therefore we assumed CCA was used. Following Harrell's guidelines,[44] CCA should

only be used if the percentage of missingness is <5% but the appropriateness of the CCA approach could not be ascertained as most of the included studies failed to report the proportion of missing data per variable. Inappropriate use of CCA results in use of only a small subset of the data which cannot be regarded as a random sample from the target population unless data are Missing Completely At Random,[45] a mechanism which is rare in practice.[15] Consequently, there are concerns about possible loss of precision in inferences and the potential biases of the estimated parameters[46] in the models employing CCA. While multiple imputation by chained equations is the principled method of imputing missing data, implementing this method when the data are not MAR could result in biased model quantities.[47] As a result, sensitivity analyses of the resultant imputations is recommended to investigate the departure from MAR assumption.[48] However, this was not the case in the studies that performed imputations on their data. Finally, handling of continuous predictors was also a concern in this review. Of the 13 models including continuous predictors, eight models[18 19 23–26 36 49] categorised continuous predictors where a continuous scale would have been possible. While this approach is intuitive to most researchers, its simplicity comes at a considerable cost of predictive performance.[50] The resulting prognostic models have been shown to have poor predictive accuracy because of the loss of statistical power and information. It is recommended that the nature of continuous data should be reserved or be handled by appropriate techniques, for example, flexible parametrisations such as fractional polynomial, regression splines or apply non-parametric techniques such as locally estimated scatterplot smoothing functions.[50 51] In this review, appropriate methods of transforming continuous data was done by only two studies[20 27] which applied restricted cubic splines and fractional polynomial.

Sixteen models attained the discrimination metric of above 80%, an indicator of promising models. However, given that the median mortality of the included studies was 6.7%, the performance reported should be interpreted with caution on account of heavily imbalanced data as a result of the rare nature of the outcome of interest. For instance, in a study with a mortality rate of 5%, a model predicting no deaths could easily attain 95% accuracy which could be potentially misleading.[33 52] Therefore, authors should report additional measures of model performance such as model specificity, sensitivity, accuracy, positive and negative predictive values for models to be contextualised appropriately.

## COMPARISON WITH OTHER STUDIES
Methods used to assess quality measures of the included models in the current study have been applied previously to critically evaluate the quality of predictive models in other specialties.[53–55] Just like the findings of this review, other previous reviews[9 56–58] describing the development of prognostic models highlighted many flaws including inappropriate statistical analyses, poor reporting of key methodological information necessary for model validation, and lack of external validations in general. Detailed and transparent reporting of the methods used in model development is one of the core principles of integrity in research because this is the only way the research community is able to evaluate study findings, and the assessment of RoB.[59] Incomplete reporting of clinical models limits future studies on prognostic research from building on the information of already existing models. This has been marked as an important source of wasted research efforts.[60] For example, external validation of prognostic models requires a full model formula to enable direct estimation of survival probabilities.[30] However, this was presented in only four models. Five models[18 19 21 23 36] that were reported to have undergone external validation did not report full model formula as required. It is, therefore, not clear whether authors of these external validation studies applied model coefficients to the external datasets, or they estimated new model coefficients (essentially model redevelopment). Thus, this review highlights the need for researchers to adhere to the TRIPOD guidelines that were created to help authors of prognostic models write complete and transparent reports. Of note, the quality of clinical predictive models does not appear to have improved over time as previous reviews from 1996,[61] 1997,[62] 2001,[63] 2005,[64] 2011,[8] 2012,[65] 2016,[66 67] 2017[68]–2019[69] have consistently identified suboptimal methodologies in the development of the score/predictive models especially in the domain of analysis. Poorly derived models may result in overoptimistic results and misleading performances. Presumably there are reasons why many prognostic models are of poor quality, including pressure to publish new predictive model regardless of the clinical value of the resultant model,[70] and inadequate biostatistical support to investigators. As observed by one of the reviewers of this study, some of the issues identified in this review such as absence of the details on the model development process can be corrected during the review and the editorial process by the journals publishing the work. There is therefore a role for editorial process for promoting best practices and recommendations of developing predictive models stated in the TRIPOD statement and ensuring compliance by authors as part of checklist for submission.

## Implications of this study
Prognostic model development workflow include development, validation (internal and external), impact assessment and implementation. Most of the included models are still in the first step of the workflow. This suggests that researchers focus more on deriving new models, often using similar prognostic factors, rather than validating and improving existing prognostic models. This leaves healthcare policy-makers with doubts as to which model to recommend in their setting. It is now time to move the prognostic research to the next step (external validation). Large patient-level datasets such as that of the Clinical

Information Network[3] which has been collected over time from a number of referral hospitals now exist in Kenya and it has been used to answer a number of salient clinical questions relevant across a range of resource-limited setting.[71–73] Future studies on prognostic research should leverage such datasets to externally validate competing models identified in this review for comparative performances as recommended by Collins and Moons[74] and if necessary, predictive performance of such models should be improved by addition of new prognostic factors. We also noted that most of the included models simplified the original predictor coefficients by rounding them to a nearest integer. This practice has an implication on model performance during external validation due to loss in predictive accuracy arising from rounding coefficients to nearest integers.[15]

We now provide guidance on methodological concerns about the candidate predictors as noted in this review. While considering potential candidate predictors to include in the prediction model, researchers should focus on the predictors that will be available at the time the prediction is made. We acknowledge that some predictors obtained from invasive procedures for example, C reactive protein, blood gas analyses, blood or cerebrospinal fluid culture, might have a higher predictive value for mortality compared with predictors derived from subjective clinical assessments, however, in resource-limited settings results of such laboratory tests typically take days to be reported or resources might not available to perform such tests in many hospitals. Consequently, models utilising such variables might not be useful to clinicians to make a decision at typical emergency departments in LMIC. Screening of model candidate predictors based on the bivariate associations whereby predictors are selected if they meet some p value threshold (commonly 0.05) have been strongly discouraged previously.[75 76] Categorising continuous model predictors is a common practice by researchers, however, this practice discards a lot of information and its assumptions are rarely clinically plausible.[33] Finally, there is a risk of overfitting if the model includes more predictors than the dataset can support. The ratio of the events (deaths) to the number of independent candidate predictors have been discussed extensively in methodological papers elsewhere[77 78] and it has been recommended that ratio of the EPV should be at least 10.

### Strengths and limitations

To our knowledge, this is the first systematic review identifying prognostic models and scoring systems predicting in-hospital all-cause paediatric mortality in low-income and middle-income settings. Our robust search strategy yielded a number of potentially eligible studies, hence it is unlikely that any potentially eligible study was not included. The quality of included models was assessed based on recent reporting standards and applied to the identified studies. For instance, if no mention was made of internal validation or even verification of the model assumptions, we could not determine whether these

crucial steps of model development were actually carried out. Thus, models that could have been otherwise rated as low RoB were rated as either unclear or high RoB in each domain. The PROBAST's analysis domain has most (9 out of 20) of the signalling questions and any given model in this domain had much higher chance to be defined as high risk as long as there was one negative (no or probably no) answer. This strict criterion led to all models being classified as either unclear or high RoB, and therefore, meta-analysis was not performed. We acknowledge that if we somewhat relaxed this decision rule, our conclusion could change. Despite this, we still hold that authors should adhere to guidelines of transparent and complete reporting of any proposed prognostic model to facilitate its external validation and subsequent application in practice. Finally, we used Google Translate to interpret a study by Bitwe et al[49] from French to English. It is possible that some statistical terminologies were not rendered correctly, or some model characteristics were lost in translation.

## CONCLUSION

Rigorously developed and robustly validated promising predictive models have the potential for improving child survival in resource-limited countries. This review identified models predicting in-hospital mortality for paediatrics. However, none of them is of good quality. Our research highlights the need to improve on the identified quality deficiencies when developing prognostic models in the future by adhering to existing generally accepted standardised methodological criteria. Majority of the derived models have not been externally validated as required. Inadequate reporting observed in the included models hinders rigorous external validation by other researchers in addition to undermining their application. Rather than developing new prognostic models, researchers should carry out comprehensive joint external validation of the identified models using large datasets ideally collected over extended time periods and different locations. This will allow model comparisons and adaptation of the competing models, if necessary, to ascertain their generalisability.

**Contributors** The roles of the contributors were as follows: ME, SA and AA conceptualised the study. MO, LM and JA conducted electronic searches to identify eligible models and did analyses. MO drafted the initial manuscript with SA, NOO, RJS, AA and ME contributed to its development. All authors read and approved the final manuscript.

**Funding** Funds from The Wellcome Trust (#207522) awarded to ME as a Senior Fellowship together with additional funds from a Wellcome Trust core grant awarded to the KEMRI-Wellcome Trust Research Programme (#092654 and #203077) supported this work. SA was supported by the Initiative to Develop African Research Leaders (IDeAL) Wellcome Trust award (#107769).

**Disclaimer** The funders had no role in drafting or submitting this manuscript.

**Map disclaimer** The depiction of boundaries on this map does not imply the expression of any opinion whatsoever on the part of BMJ (or any member of its group) concerning the legal status of any country, territory, jurisdiction or area or

of its authorities. This map is provided without any warranty of any kind, either express or implied.

**Competing interests** None declared.

**Patient and public involvement** Patients and/or the public were not involved in the design, or conduct, or reporting, or dissemination plans of this research.

**Patient consent for publication** Not required.

**Provenance and peer review** Not commissioned; externally peer reviewed.

**Data availability statement** All data relevant to the study are included in the article or uploaded as online supplemental information.

**ORCID iD**
Morris Ogero http://orcid.org/0000-0003-0117-6289

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
