## [Reviewer comments · BMJ Open]

ARTICLE DETAILS

TITLE (PROVISIONAL)	Prognostic models for predicting in-hospital paediatric mortality in resource-limited countries: a systematic review
AUTHORS	Ogero, Morris; Sarguta, Rachel; Malla, Lucas; Aluvaala, Jalemba; Agweyu, Ambrose; English, Mike; Onyango, Nelson; Akech, Samuel

VERSION 1 – REVIEW

REVIEWER	Asya Agulnik St. Jude Children's Research Hospital USA
REVIEW RETURNED	27-Dec-2019

GENERAL COMMENTS	The authors present a well-written manuscript describing a systematic review of prognostic models for inpatient pediatric mortality in resource-limited countries. The study is rigorously conducted, and I appreciate the author's focus on the methodological and statistical issues in the currently available studies. I do have a few suggestions for improvement: - I appreciate the author's emphasis on the need for external validation of existing predictive models. It would be helpful to hear how the models that have had external validation were done, pointing out any issues with these approaches, as guidance on how this can be done in the future.- While I understand that the author's analysis shows that no current model is totally free of risk of bias, it would be helpful for the authors to describe the model(s) that are the best in the existing literature (from figure 5 it looks like maybe Lowlaavar model 3), and spend some time talking about the elements of that model. More discussion of the models themselves, beyond their analytical flaws, would make the paper more clinically relevant overall for readers. What is the current best model(s) from the literature as judged by the authors from their analysis?- I think the author's overall conclusion—that published models have a high risk of bias due primarily to bias in the analysis—is important. I would like to hear more about why the authors think this is the case in the discussion? Is this an issue with resources for adequate biostatistical support? Or possibly issues with the review process in the journals publishing the work (much of what the authors point out is a lack of describing parts of the development/analysis process rather than methodological flaws, which could be corrected in review). Or are there other potential reasons? Discussing these is important as a step towards correcting the issues that exist in the current literature.
--

	- I would like to see more guidance from the authors re: what needs to happen in future work (expansion on the section of “implication of this study”. Here the authors focus on the need for external validation, but it seems that there is more guidance that can be provided based on common flaws in the current literature. - The supplementary file 2 is not formatted appropriately making it impossible to read all the data in the cells. I think the formatting/font can be adjusted to make this more legible
--	---

REVIEWER	Christopher Horvat and Jonathan Pelletier (PCCM Fellow) UPMC Children's Hospital of Pittsburgh Pittsburgh, Pennsylvania, United States of America
REVIEW RETURNED	08-Mar-2020

GENERAL COMMENTS	Summary: Dr. Ogero et. al. performed a systematic review of mortality prediction models specifically applied to patients with common pediatric diseases in low- and middle-income countries. Identifying robust, well-performing models in LMICs is important as such models are important for risk adjusting analyses in this unique population. They found 15 articles discussing 21 models, and overall found a poor quality of statistical analysis and results reporting. They were unable to perform meta-analysis due to study heterogeneity. By the authors account, theirs is the first such analysis in this specific population. Their findings highlight the need for standardization of model development and reporting and they make an appropriate call to use larger, multi-center data collections for future model development and to perform external validation. I think this work would be a valuable addition to an increasing body of literature focusing on the development of predictive models; however, there are several major and minor issues that should be addressed. Major Comments:  • The findings in Figure 5 and supplementary file 3 are discordant with respect to the models developed by Lowlaavar 2016. These models are marked as having unclear risk of analytic bias in the figure, but low risk of analytic bias in the supplementary file. This is especially relevant given that they are the only 3 models that would be given this status. Please clarify. • Many of the tables contain cells that are cut-off and cannot be read. I cannot interpret much of supplementary file 2 for this reason. • Most of the models are being derived on heavily imbalanced data sets (the authors cite a median mortality of only 6.7%). In the results, page 11, the authors cite an AUROC of 0.8 as a “promising” model. However, this method of reporting discrimination is likely falsely elevated in the face of such low mortality. The authors should add additional discussion / recommendations that additional measures of discrimination (e.g. precision/recall or average precision) be reported. • Methods, page 8: “if a predictive model was rated as low RoB for all domains and it has not been subjected to any external validation, we downgraded it to high RoB.”  o Given that nearly every model in this study was rated as high RoB, it would be useful to distinguish which models are internally consistent but require additional validation studies from those that are internally flawed. This is particularly important given that the
---

	authors specifically call for external validation studies in their “implications of this study” on page 16. □ This could be accomplished by a separate category such as “low RoB, but lacking external validation.” Minor Comments:  • Methods, page 6: I have some concerns about the use of Google Translate for this type of technical systematic review. It is highly plausible that Google Translate would render errors in statistical terminology. Given that the present authors did not contact primary study authors for additional data or clarification, it is entirely possible that some studies reported additional model characteristics that were lost in translation.  o I would request that the authors explicitly state for which manuscripts interpreter services were used and cite this as a limitation. • Results, page 9: “Five reports of the 15 included studies utilized data from at least two centres of which 3 studies 21 were conducted in multiple countries including sub-Saharan Africa and Asian countries (Figure 2).”  o Please clarify this statement. Figure 2 lists the 3 multiple-country studies, but no studies in only 2 countries. • Discussion, page 14: “Inappropriate use of CCA results in use of only a small subset of the data which cannot be regarded as a random sample from the target population unless data are missing completely at random (MCAR), a mechanism which is rarely in practice.”  o I agree with the authors that medical data are very rarely missing at random. This leads to problems with data imputation by any method. The underlying assumption of MICE imputation is that missing data is missing at random after controlling for variables that are present in the model (see PMID: 21499542). It is likely that no single imputation method is the “best” for medical data, and that the “best” approach is to perform sensitivity analyses with a variety of different imputation methods and compare results. • Figure 2 legend: “Text highlighted in red are the names of the models with their corresponding discrimination measures.”  o Please clarify which discrimination measure is displayed here. Is this an AUROC? AUPRC? Given a median mortality of only 6.7% (heavily imbalanced data set), AUROC is likely not the best measure of discrimination (at least if reported in isolation). • Page 15, line 19: replace “a lot of” with “many” • It would be worth noting the recently published guidance on how to report the development and performance of predictive models (doi: 10.1097/CCM.0000000000004246)
--	--

REVIEWER	Tzy-Jyun Yao Harvard T.H.Chan School of Public Health USA
REVIEW RETURNED	19-Apr-2020

GENERAL COMMENTS	This manuscript is well written. Although unfortunately, no meta-analysis was performed, it is justified given all the identified prognostic models had high risk of bias. I think the finding is alarming and worth reporting. A few changes/clarifications should be made before publication:
---

	 1. It is not clear how risk of bias in analysis was determined. In pages 44-46, models RISC HIV+ and RISC-Malawi had 1 No and 2 No information out of 9 questions and were assigned as High risk, while Lowlaavar Models 1, 2 and 3 also had 1 No and 2 No information but were assigned as Low risk. Nonetheless, the listed risk levels on page 46 were not consistent with what reported in Figure 5. 2. The number of signaling questions of the 4 domains ranged from 2 to 9. The authors never spelled out the decision rules for high risk, but seemed to define a domain as high risk as long as there was one negative answer within that domain. This practice made domains with more signal questions have much higher chance to be defined as high risk, particularly for the domain of analysis, which led to high risk for all studies and further analysis was not attempted. Somewhat relaxed decision rule may be worth considering, and so meta-analysis could be performed on the subset of lower risk models, such as RISC HIV+ and Lowlaavar Model 3. 3. The authors referred to several previous reviews and pointed out the quality has not been improved. More elaborations of the change or not change in different aspects or domains of quality would have been useful. 4. Limitations of this review were not discussed. An example such as described above in item 2 is a limitation of this review. 5. A typing error on page 8, "...model was judged as low on all five domains,..." should be four domains. The situation of missing data should be reported and explored whether the missing had any pattern. If some pattern did exist, they should be addressed in the analysis. The key finding of this study seemed to be that the scores were very low across all groups. All the mean scores were lower than the mid-value of 3.5 (between 1 to 6). I think this should be the focus of the discussion.
--	--

VERSION 1 – AUTHOR RESPONSE

Reviewer(s)' Comments to Author:

Reviewer: 1

The authors present a well-written manuscript describing a systematic review of prognostic models for inpatient pediatric mortality in resource-limited countries. The study is rigorously conducted, and I appreciate the author's focus on the methodological and statistical issues in the currently available studies. I do have a few suggestions for improvement:

- I appreciate the author's emphasis on the need for external validation of existing predictive models. It would be helpful to hear how the models that have had external validation were done, pointing out any issues with these approaches, as guidance on how this can be done in the future.

Response: I agree it would have been very useful to critique externally validated models to guide future studies. While this suggestion is beyond the scope of this review, we have added in the revised manuscript in the section titled “comparison with other studies” a note questioning the procedure of how the five externally validated models was done considering that these models did not report a full model formula (including coefficients and intercept) as required. It is therefore not clear whether authors of these external validation studies applied model coefficients to the external datasets, or they estimated model coefficients (essentially model redevelopment).

- While I understand that the author’s analysis shows that no current model is totally free of risk of bias, it would be helpful for the authors to describe the model(s) that are the best in the existing literature (from figure 5 it looks like maybe Lowlaavar model 3), and spend some time talking about the elements of that model. More discussion of the models themselves, beyond their analytical flaws, would make the paper more clinically relevant overall for readers. What is the current best model(s) from the literature as judged by the authors from their analysis?

Response: all models had high or unclear RoB in the statistics/analysis domain, however in considering the other 3 domains the models with low RoB in the remaining 3 domains would have been considered as the “best” models. However, we felt that individual domains should not be interpreted separately instead the ratings from all domains should be used to determine model’s overall judgement of the risk of bias as recommended. Moreover, there is no empirical evidence which is most important source of bias <https://www.acpjournals.org/doi/10.7326/M18-1376>

- I think the author’s overall conclusion—that published models have a high risk of bias due primarily to bias in the analysis—is important. I would like to hear more about why the authors think this is the case in the discussion? Is this an issue with resources for adequate biostatistical support? Or possibly issues with the review process in the journals publishing the work (much of what the authors point out is a lack of describing parts of the development/analysis process rather than methodological flaws, which could be corrected in review). Or are there other potential reasons? Discussing these is important as a step towards correcting the issues that exist in the current literature.

Response: We agree with the reviewer’s observation. We have added these ideas to the fifth paragraph of the discussion in revised the manuscript.

- I would like to see more guidance from the authors re: what needs to happen in future work (expansion on the section of “implication of this study”). Here the authors focus on the need for external validation, but it seems that there is more guidance that can be provided based on common flaws in the current literature.

Response:

In the revised manuscript we have highlighted the pipeline of prognostic model development which include development, validation (internal and external), impact assessment and implementation. Most of the included models are still in the first step of the pipeline. This suggests that researchers focus more on deriving new models. It is now time to move the prognostic research to the subsequent steps.

We have also added a paragraph in the discussion providing guidance on the common flaws identified in this review and how these can be addressed

- The supplementary file 2 is not formatted appropriately making it impossible to read all the data in the cells. I think the formatting/font can be adjusted to make this more legible

Response: We apologise for this. This is because the supplementary files originally in MS Excel, owing to their width were distorted when converted to PDF during submission to the journal. We have now formatted them accordingly including reducing font sizes to make it fit within A4 pages.

Reviewer: 2

Please leave your comments for the authors below
Summary:

Dr. Ogero et. al. performed a systematic review of mortality prediction models specifically applied to patients with common pediatric diseases in low- and middle-income countries. Identifying robust, well-performing models in LMICs is important as such models are important for risk adjusting analyses in this unique population. They found 15 articles discussing 21 models, and overall found a poor quality of statistical analysis and results reporting. They were unable to perform meta-analysis due to study heterogeneity. By the authors account, theirs is the first such analysis in this specific population. Their findings highlight the need for standardization of model development and reporting and they make an appropriate call to use larger, multi-center data collections for future model development and to perform external validation. I think this work would be a valuable addition to an increasing body of literature focusing on the development of predictive models; however, there are several major and minor issues that should be addressed.

Major Comments:

- The findings in Figure 5 and supplementary file 3 are discordant with respect to the models developed by Lowlaavar 2016. These models are marked as having unclear risk of analytic bias in the figure, but low risk of analytic bias in the supplementary file. This is especially relevant given that they are the only 3 models that would be given this status. Please clarify.

Response: *This was an error in our part. We have since rectified it in the revised manuscript*

- Many of the tables contain cells that are cut-off and cannot be read. I cannot interpret much of supplementary file 2 for this reason.

Response: *We apologise for this. This is because the supplementary files originally in MS Excel, owing to their width were distorted when converted to PDF during submission to the journal. We have now formatted them accordingly including reducing font sizes to make them fit A4 pages.*

- Most of the models are being derived on heavily imbalanced data sets (the authors cite a median mortality of only 6.7%). In the results, page 11, the authors cite an AUROC of 0.8 as a “promising” model. However, this method of reporting discrimination is likely falsely elevated in the face of such low mortality. The authors should add additional discussion / recommendations that additional measures of discrimination (e.g. precision/recall or average precision) be reported.

Response:

We have added this in paragraph 6 of the discussion in the revised manuscript

It is also worth noting that such results are expected given the sample size – hence the high risk of bias in achieving this result

- Methods, page 8: “if a predictive model was rated as low RoB for all domains and it has not been subjected to any external validation, we downgraded it to high RoB.”

o Given that nearly every model in this study was rated as high RoB, it would be useful to distinguish which models are internally consistent but require additional validation studies from those that are internally flawed. This is particularly important given that the authors specifically call for external validation studies in their “implications of this study” on page 16.

□ This could be accomplished by a separate category such as “low RoB, but lacking external validation.”

Response: *We agree with this suggestion; however, no model was downgraded from Low RoB to High Rob since all models were either high or unclear RoB and we have added this suggestion to the revised manuscript in the discussion section titled ‘Summary of key findings’. PROBAST tool assesses internal validity and the RoB statement is a summary measure of the internal validity.*

Minor Comments:

- Methods, page 6: I have some concerns about the use of Google Translate for this type of technical systematic review. It is highly plausible that Google Translate would render errors in statistical terminology. Given that the present authors did not contact primary study authors for additional data or clarification, it is entirely possible that some studies reported additional model characteristics that were lost in translation.

- I would request that the authors explicitly state for which manuscripts interpreter services were used and cite this as a limitation.

Response: We appreciate this concern and we have cited a study where we used Google Translate to translate text from French to English. We have also added in the limitation that some information was potentially lost during translation.

- Results, page 9: “Five reports of the 15 included studies utilized data from at least two centres of which 3 studies were conducted in multiple countries including sub-Saharan Africa and Asian countries (Figure 2).”

- Please clarify this statement. Figure 2 lists the 3 multiple-country studies, but no studies in only 2 countries.

Response: We have reworded this statement in the revised manuscript

- Discussion, page 14: “Inappropriate use of CCA results in use of only a small subset of the data which cannot be regarded as a random sample from the target population unless data are missing completely at random (MCAR), a mechanism which is rarely in practice.”

- I agree with the authors that medical data are very rarely missing at random. This leads to problems with data imputation by any method. The underlying assumption of MICE imputation is that missing data is missing at random after controlling for variables that are present in the model (see PMID: 21499542). It is likely that no single imputation method is the “best” for medical data, and that the “best” approach is to perform sensitivity analyses with a variety of different imputation methods and compare results.

Response: The reviewer is right that sensitivity analyses should be performed in different scenarios after the multiple imputation to assess the departure of missing at random assumption. In the revised manuscript we have captured these suggestions in the discussion under paragraph 4.

- Figure 2 legend: “Text highlighted in red are the names of the models with their corresponding discrimination measures.”

- Please clarify which discrimination measure is displayed here. Is this an AUROC? AUPRC? Given a median mortality of only 6.7% (heavily imbalanced data set), AUROC is likely not the best measure of discrimination (at least if reported in isolation).

Response: We have clarified in the legend of the revised manuscript that the discrimination metric is AUROC. We have also added in the discussion section that AUROC derived from imbalanced data should be interpreted with caution. We have also discussed that additional performance metrics such as positive/negative predictive values should also be reported.

- Page 15, line 19: replace “a lot of” with “many”

Response: Revised line 19 where we replaced “a lot of” with “many”

- It would be worth noting the recently published guidance on how to report the development and performance of predictive models (doi: 10.1097/CCM.0000000000004246)

Response: We appreciate this reference material and we have used to revise our manuscript accordingly and cited it appropriately.

Reviewer 3

This manuscript is well written. Although unfortunately, no meta-analysis was performed, it is justified given all the identified prognostic models had high risk of bias. I think the finding is alarming and worth reporting. A few changes/clarifications should be made before publication:

1. It is not clear how risk of bias in analysis was determined. In pages 44-46, models RISC HIV+ and RISC-Malawi had 1 No and 2 No information out of 9 questions and were assigned as High risk, while Lowlaavar Models 1, 2 and 3 also had 1 No and 2 No information but were assigned as Low risk. Nonetheless, the listed risk levels on page 46 were not consistent with what reported in Figure 5.

Response: This was a typographical error in our part, we have revised the manuscript and corrected the inconsistencies.

2. The number of signaling questions of the 4 domains ranged from 2 to 9. The authors never spelled out the decision rules for high risk, but seemed to define a domain as high risk as long as there was one negative answer within that domain. This practice made domains with more signal questions have much higher chance to be defined as high risk, particularly for the domain of analysis, which led to high risk for all studies and further analysis was not attempted. Somewhat relaxed decision rule may be worth considering, and so meta-analysis could be performed on the subset of lower risk models, such as RISC HIV+ and Lowlaavar Model 3.

Response: We agree with the reviewer that the analysis domain of PROBAST contain many items and therefore any given model has a comparatively higher propensity to be scored “high RoB”. We also acknowledge that our decision rule was strict. However, we decided not to modify the decision rules nor reduce the number of the PROBAST items used in determining the risk of bias because we believe that authors should adhere to guidelines of transparent and complete reporting of any proposed prognostic model. Risk of bias can only be assessed based on available data as reported by authors of the eligible studies

<https://onlinelibrary.wiley.com/doi/epdf/10.1002/hsr2.165>

However, in the revised manuscript we have added that our strict decision rule is a potential limitation of the study.

3. The authors referred to several previous reviews and pointed out the quality has not been improved. More elaborations of the change or not change in different aspects or domains of quality would have been useful.

Response: *In the revised manuscript we have added that the quality of the analysis domain has not improved over time.*

4. Limitations of this review were not discussed. An example such as described above in item 2 is a limitation of this review.

Response: *This has been added as limitation as explained in our response to item 2.*

5. A typing error on page 8, “...model was judged as low on all **five** domains,...” should be **four** domains.

Response: *This has been noted and changed from five to four in the revised manuscript.*

The situation of missing data should be reported and explored whether the missing had any pattern. If some pattern did exist, they should be addressed in the analysis.

The key finding of this study seemed to be that the scores were very low across all groups. All the mean scores were lower than the mid-value of 3.5 (between 1 to 6). I think this should be the focus of the discussion.

Response: *We appreciate this concern however we had observed in the manuscript that the authors of the included studies rarely reported the extent of missing data and how it was handled. Therefore, it was not possible to discern patterns.*

VERSION 2 – REVIEW

REVIEWER	Chris Horvat and Jonathan Pelletier UPMC Children's Hospital of Pittsburgh
REVIEW RETURNED	13-Aug-2020

GENERAL COMMENTS	The authors have provided a satisfactory response to our initial critique.
--

REVIEWER	Tzy-Jyun Yao Harvard T.H. Chan School of Public Health, Boston, USA
REVIEW RETURNED	26-Jul-2020

GENERAL COMMENTS	The authors addressed almost all of my comments and have largely improved the discussion. There is only one confusing sentence, the last sentence on page 12: “Two models were judged to be high RoB because at least one of the domains was rated high RoB”. This sentence might be wrong because just two sentences before, it was clearly and accurately stated “9 out of 21 models were judged to be of high risk of bias because at least one out of four domains in these models were rated high RoB
--

VERSION 2 – AUTHOR RESPONSE

Reviewer(s)' Comments to Author:

Reviewer: 3

Reviewer Name: Tzy-Jyun Yao

Institution and Country: Harvard T.H. Chan School of Public Health, Boston, USA Please state any competing interests or state 'None declared': None declared.

Please leave your comments for the authors below.

1. The authors addressed almost all of my comments and have largely improved the discussion. There is only one confusing sentence, the last sentence on page 12: "Two models were judged to be high RoB because at least one of the domains was rated high RoB". This sentence might be wrong because just two sentences before, it was clearly and accurately stated "9 out of 21 models were judged to be of high risk of bias because at least one out of four domains in these models were rated high RoB."

Response: *In the revised manuscript we have deleted the confusing sentence as highlighted by the reviewer*

Reviewer: 2

Reviewer Name: Chris Horvat and Jonathan Pelletier Institution and Country: UPMC Children's Hospital of Pittsburgh Please state any competing interests or state 'None declared': None

Please leave your comments for the authors below

The authors have provided a satisfactory response to our initial critique.